# Water-Use Efficiency of Crops in the Arid Area of the Middle Reaches of the Heihe River: Taking Zhangye City as an Example

**Xingyuan Xiao** [1,2,*]**, Limeng Fan** [1]**, Xiubin Li** [2]**, Minghong Tan** [2]**, Tao Jiang** [1]**, Luqian Zheng** [2] **and Fengrui Jiang** [3]

1 College of Geomatics, Shandong University of Science and Technology, Qingdao 266590, China
2 Key Laboratory of Land Surface Pattern and Simulation, Institute of Geographic Sciences and Natural Resources Research, Chinese Academy of Sciences, Beijing 100101, China
3 Weifang Middle School, Weifang 261041, China
* Correspondence: xiaoxy_111@163.com; Tel.: +86-13553095709

**Abstract:** The middle reaches of the Heihe River are an important food base in the arid regions of Northwest China. The agricultural water consumption in this region accounts for about 90% of the total water consumption. The shortage of water resources is the primary reason for restricting agricultural development. Therefore, studying the efficiency of agricultural water use is essential to improving the effective use of water resources. Under the premise of considering agricultural water saving, we improved the water efficiency model from the perspective of pure agricultural income that farmers are more concerned about. In this study, we took Zhangye City in the middle reaches of the Heihe River as an example, based on meteorological crop data and farmer survey data. Then, we used the input–output method to quantitatively analyze the net income of the crops in Zhangye City. We used the CROPWAT model to calculate the water demand of crops during the growing season. Lastly, we used the improved water-use efficiency (WUE) model to analyze WUE differences of crops in the study area. We reached the following conclusions: (1) among the six crops in the study area, the net profit of seed corn was 20,520 yuan/ha, followed by field corn, 11,700 yuan/ha, then followed by potato, rapeseed, wheat, and barley; (2) the maximum water requirement for the crop growth period was 597.2 mm for field corn, followed by 577.3 mm for seed corn, then followed by rapeseed, wheat, barley, and potato; (3) among the six crops, the WUE calculated using the water efficiency model before and after improvement had obvious differences. The WUE calculated using the original model reached 9.03 yuan/m$^3$ for potato, followed by 6.33 yuan/m$^3$ for seed corn. The WUE calculated using the improved model reached 3.44 yuan/m$^3$ for seed corn, which is the maximum, followed by potato with 2.25 yuan/m$^3$. Considering the agricultural water saving and crop yields, we propose to properly expand the cultivation of seed corn and potato in the middle reaches of the Heihe River. This would be more conducive to achieving a "win-win" situation for water conservation and revenue.

**Keywords:** water-use efficiency; CROPWAT model; input–output model; Zhangye City

## 1. Introduction

Water is an irreplaceable resource for human survival and development and the basis for sustainable economic and social development [1]. With the rapid population growth and rapid socio-economic development, as well as the impact of global climate change, water shortages have become more serious and one of the common concerns of the world [2,3]. In the report of the 19th National Congress of the Communist Party of China, it was clearly stated that "promoting all-round conservation and recycling of resources and implementing national water conservation actions" is

necessary [4]. From the perspective of water-use structure, agriculture is the most important water sector, accounting for 62% of total water consumption [5], while especially in the northwestern arid regions, agricultural water accounts for about 90% of total water consumption [6]. Therefore, the development of water-saving agriculture and planting water-saving crops have an important theoretical and practical significance for improving agricultural water-use efficiency in arid areas, reducing water waste, and promoting local economic development.

The Heihe River Basin is located in the arid northwestern region of China. The area is an important grain production base in China. The annual agricultural irrigation water consumption is about 1.597 billion m$^3$, of which 33.6% comes from groundwater [7]. The over-occupation of water resources by agricultural irrigation makes the groundwater depth in the Heihe River Valley decrease at a rate of 0.5–1.8 m per year [8,9], which causes eco-environmental problems such as shrinkage of downstream natural oases, decline in vegetation coverage, and desertification of land [10,11], between upstream agricultural water use and downstream ecological water demand. The contradiction is prominent [12]. Therefore, agricultural water saving in the middle reaches of the Heihe River is an important starting point for the rational use of Heihe water resources.

At present, agricultural water saving research in arid areas is mostly carried out in terms of agricultural water-use efficiency, where agricultural water is used during the period of crop growth using water sources (natural precipitation and irrigation water), for water distribution (water conservancy projects, channels), and in the field (crop evapotranspiration, crop yields). There are water loss problems in every aspect of water distribution channels, fields, and crops. Therefore, agricultural water efficiency can be divided into water distribution efficiency, field water efficiency, and crop water-use efficiency [13].

After the 1970s, from the perspective of crop physiology, the academic community defined agricultural water-use efficiency as water-use efficiency (WUE), which refers to the economic output of crops produced by unit water consumption (expressed as WUE = crop yield/water consumption) and is an indicator of the relationship between crop yield and water use [14]. There are three types of water use here. One is the water consumption of the crop, the evapotranspiration ($ET_c$), which is the water consumption characteristic of different crops. The second is the amount of irrigation water. The third is the natural precipitation. Based on the water consumption of crops, Hu et al. measured the water-use efficiency of wheat in Minle County and corn in Linze County, and studied the influencing factors [15,16]. Tan and Zheng used the CROPWAT model to simulate the corn production in the Heihe River Basin. The results show that there are significant differences in the water consumption characteristics of the two types of corns during the growing season [17]. Zhang et al. quantitatively calculated the water-use efficiency of dryland corn in the Loess Plateau of China, and based on this, explored methods for improving water-use efficiency [18]. Based on the amount of irrigation water, Liu et al. used remote sensing technology and the Penman–Monteith (PM) formula to calculate the temporal and spatial variation characteristics of the irrigation water demand for different crop types in the middle reaches of the Heihe River in 2007 and 2012 [19]. Zheng et al. [20] compared with the irrigation water efficiency of Chinese and American crops on a regional scale. The water-use efficiency based on natural precipitation is mainly used to indicate the water conversion efficiency of dryland [21].

In the above study, the calculation of crop WUE used the ratio of crop yield to water use, which we call the original model. In the original model, the calculation of water efficiency only considered the crop yield, and did not consider the difference between crop yield and input.

At the regional level, there are many types of crops planted, and the economic yields of various crops are quite different. For example, the economic yield of wheat is grain yield, whereas that of potato is tuber yield. It is therefore difficult to determine the average WUE. In this regard, Duan [21] proposed in 2005 that economic products should be converted according to their market value. In addition, the economic benefit of the unit water consumption should be used to express the WUE of the crop. The WUE expressed by this method is also called the water-use benefit (expressed as WUE = crop yield × crop market price/water consumption). According to this definition, Zheng and Tan [22] studied the WUE of barley, wheat, corn, and rape in the middle reaches of the Heihe River, and proposed

suggestions for the adjustment of the agricultural structure in the region. Using this definition, crops with large differences in yield can be unified into market value at the market price, and then the average WUE of crops in a region can be calculated to compare the differences in WUE between different crops. However, the stability of this conversion method is poor, and it will change greatly with a change in the market value of agricultural products. At the same time, the final yield of crops may be related to some external inputs such as the seed quality of crops, the amount of pesticides and fertilizers used, the use of plastic film, and the amount of mechanical labor. Considering only crop yields is not comprehensive and, accordingly, the WUE of crops calculated on this basis is also not reasonable enough.

Therefore, in this study we started from the perspective of pure agricultural income that farmers are more concerned about. We improved the water efficiency model to study the difference in WUE of crops in Zhangye City. In this way, we considered not only the agricultural water saving but also the income of the farmers. Under the premise of balancing the two factors, it is more reasonable to adjust the crop planting structure in an arid area for achieving a "win-win" situation of water conservation and revenue.

## 2. Study Area and Data

### 2.1. Overview of the Study Area

Zhangye City is located at 97°20′–102°12′east longitude and 37°28′–39°57′ north latitude, which is a typical desert oasis area (Figure 1). The precipitation is scarce, the sunshine is sufficient, and the irrigation conditions are good. Zhangye City has more than 80% of artificial oasis and 95% of cultivated land [23] in the Heihe River Basin. The water consumption accounts for 82.6% of the total water consumption in the Heihe River Basin. It is an important commodity grain production base and vegetable production base in Gansu Province, and also produces the top ten commodity grains in China [24]. The agricultural districts under its jurisdiction include Gaotai County, Linze County, Ganzhou District, Minle County, and Shandan County. The agriculture is mostly single-season planting, mainly in summer harvest and autumn harvest. The summer harvest crops are mainly wheat and barley. The autumn harvest crops mainly include corn, rapeseed and potato, and the crop growth period is mostly concentrated between April and September [25].

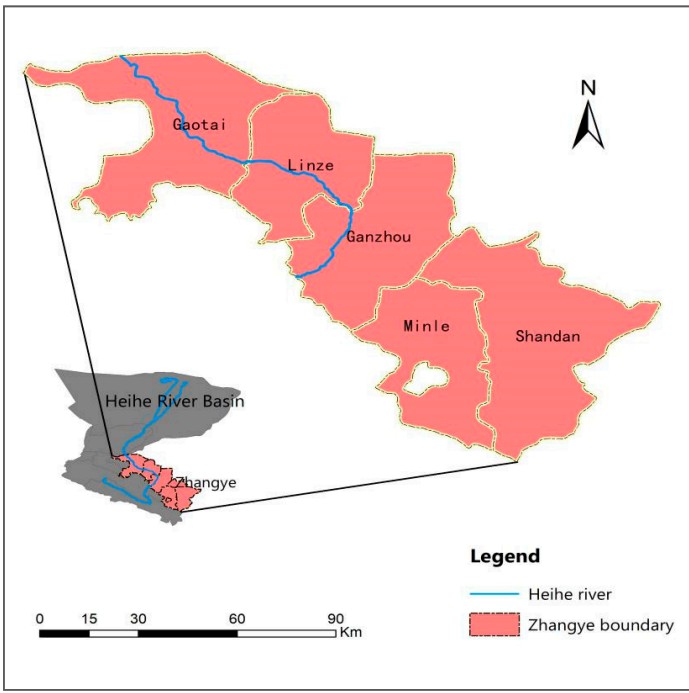

**Figure 1.** Schematic diagram of the study area.

### 2.2. Research Data

Meteorological data were obtained from the China Meteorological Data Sharing Service Network. This data set provides daily weather observation data for Zhangye Site (38°56′ N, 100°26′ E) required for this study, including daily average minimum and maximum temperatures (°C), daily average relative humidity (%) and daily average minimum relative humidity (%), daily average wind speed (m/s), sunshine hours (h), and daily precipitation (mm). Table 1 shows the monthly average data of the daily weather parameters of Zhangye Site in 2012.

**Table 1.** Monthly average of meteorological parameters of Zhangye Site in 2012.

| Parameter | January | February | March | April | May | June | July | August | September | October | November | December |
|---|---|---|---|---|---|---|---|---|---|---|---|---|
| Average minimum temperature (°C) | −19.7 | −14.6 | −4.2 | 3.7 | 10.3 | 15.2 | 17 | 15.2 | 8.4 | −0.2 | −8.8 | −14.3 |
| Average maximum temperature (°C) | −3.5 | 2.7 | 11.1 | 19.6 | 25.6 | 29.4 | 29.5 | 28.9 | 24.0 | 16.9 | 5.8 | −0.8 |
| Average relative humidity (%) | 57 | 39 | 37 | 30 | 33 | 44 | 58 | 53 | 49 | 45 | 59 | 68 |
| Average minimum relative humidity (%) | 35 | 21 | 16 | 13 | 16 | 21 | 30 | 27 | 21 | 17 | 27 | 41 |
| Average wind speed (m/s) | 1.7 | 1.8 | 2.5 | 2.5 | 2.2 | 2.2 | 2.2 | 2.1 | 1.8 | 1.9 | 2.2 | 2.0 |
| Sunshine hours (h) | 7.4 | 6.9 | 8.3 | 9.8 | 9.6 | 10.8 | 9.2 | 8.9 | 10.1 | 9.1 | 7.3 | 5.8 |
| Precipitation (mm) | 1.1 | 0.6 | 1.9 | 2.8 | 13.8 | 55.7 | 32.8 | 10.6 | 8.0 | 0.7 | 3.6 | 5.2 |

In July 2015, we conducted field visits to 5 districts and counties, 18 townships, and 27 villages in Gaotai County, Linze County, Ganzhou District, Minle County, and Shandan County under the jurisdiction of Zhangye City. At the same time, the households were randomly selected to conduct questionnaires and interviews. Out of 144 questionnaires collected from farmers, 136 were valid and the effective rate was 94.4%. The survey included the growth cycle, growth stage, root depth, plant height, and other characteristics of wheat, barley, seed corn, field corn, rapeseed, and potato. The main distribution points of the six crops are shown in Figure 2. Input and output of crops included seeds, plastic film, pesticides, fertilizers, machinery, labor, irrigation, and other costs.

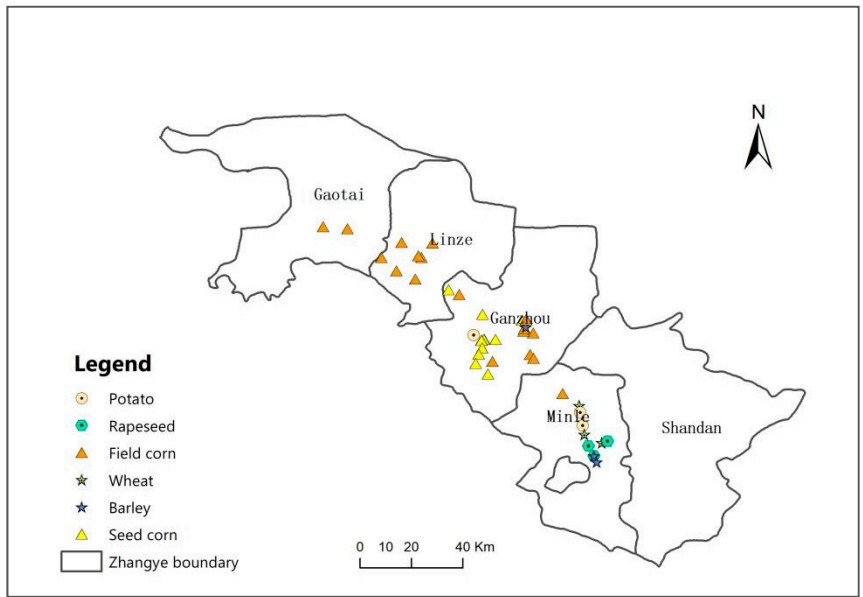

**Figure 2.** Distribution of crops in study area.

## 3. Methods

### 3.1. Input–Output Analysis

The input–output analysis, also known as the "industry linkage" analysis or the "sectoral balance" analysis, was first proposed by the American economist Wassily Leontief. It mainly reflects the interconnection between various departments (industries) of the economic system by compiling input–output tables and establishing corresponding mathematical models. The input–output table is divided into a value-based input–output table and a physical-type input–output table. The value-based input–output table describes the value flow process of each department's products, using currency as the unit of measurement. Because it can not only reflect the physical movement process of each department, but also describe the value flow process of each department's products, it is more practical [26]. Since the 1960s, geographers have applied this method extensively in resource utilization and environmental protection analysis, regional industrial composition analysis, and agriculture [27]. This paper applies this method to compile value input–output tables, and calculate the input–output of each crop to obtain crop yield and net income.

Table 2 shows a simplified value input–output table because it is measured in currency (usually with comparable prices), so that mathematical models can be built either by row or by column. The model is built according to the column, reflecting the value formation process of each department's products, reflecting the balance between production and consumption. Therefore, the input and output of various crops in this study were calculated using a model established by column. The model is as follows:

**Table 2.** Value input and output table.

| Input \ Output | | Intermediate Use | | | | | Final Product | Gross Output Value |
|---|---|---|---|---|---|---|---|---|
| | | Dep.1 | Dep.2 | ... Dep.n | Subtotal | | | |
| **Material consumption** | Dep.1 | $x_{11}$ | $x_{12}$ | ... $x_{1n}$ | $e_1$ | | $y_1$ | $x_1$ |
| | Dep.2 | $x_{21}$ | $x_{22}$ | ... $x_{2n}$ | $e_2$ | | $y_2$ | $x_2$ |
| | ... | $\vdots$ | $\vdots$ | $\vdots$ $\vdots$ | | | $\vdots$ | $\vdots$ |
| | Dep.n | $x_{n1}$ | $x_{n2}$ | ... $x_{nn}$ | $e_n$ | | $y_n$ | $x_n$ |
| | Subtotal | $c_1$ | $c_2$ | ... $c_n$ | $c$ | | $y$ | $x$ |
| **New value creation** | Labor remuneration | $v_1$ | $v_2$ | ... $v_n$ | $v$ | | | |
| | Net income | $m_1$ | $m_2$ | ... $m_n$ | $m$ | | | |
| | Subtotal | $N_1$ | $N_2$ | ... $N_n$ | $N$ | | | |
| **Gross output value** | | $x_1$ | $x_2$ | ... $x_n$ | $x$ | | | |

For *n* departments, create **n** equations:

$$
\begin{aligned}
x_{11} + x_{21} + \ldots + x_{n1} + v_1 + m_1 &= x_1 \\
x_{12} + x_{22} + \ldots + x_{n2} + v_2 + m_2 &= x_2 \\
\vdots \quad\quad \vdots \quad\quad\quad \vdots \quad\quad \vdots \quad\quad \vdots \quad\quad &\vdots \\
x_{1n} + x_{2n} + \ldots + x_{nn} + v_n + m_n &= x_n
\end{aligned}
$$

The above equation can also be written as:

$$
\sum_{i=1}^{n} x_{ij} + v_j + m_j = x_j \quad (j = 1, \quad 2, \quad \ldots, n). \tag{1}
$$

Equation (1) is also called the cost-balance equation, which reflects the relationship between material consumption costs, newly created value, and total product value. Using this model, the various input costs and output of the crops in Zhangye City can be calculated.

### 3.2. Reference Crop Evapotranspiration—Crop Coefficient Method

The key to calculating the WUE of crops is to determine the water demand of crops. In domestic and foreign research, the $ET_c$ of crops is usually used to characterize crop water demand, which indicates the actual amount of water used for crop growth. At present, the more common methods for obtaining $ET_c$ can be divided into actual measurement method, reference crop evapotranspiration–crop coefficient method, and remote sensing simulation method [28].

In recent years, the use of the reference crop evapotranspiration–crop coefficient method to estimate crop $ET_c$ has been increasingly used [29,30]. In this paper, the weather data of Zhangye Site in 2012 is substituted into the Penman–Monteith (PM) formula to calculate the reference crop evapotranspiration ($ET_0$), and then multiplied by the crop coefficient ($K_c$) to obtain the actual $ET_c$ of crops.

#### 3.2.1. Calculation of the $ET_0$

The calculation of reference crop $ET_c$ in this study uses the CROPWAT model developed by the Department of Land and Water Resources Development of the Food and Agriculture Organization (FAO). The model calculates $ET_0$ using the corrected PM formula, which is currently the only standard method recommended for calculating $ET_0$. The Irrigation Water Requirements Committee of the American Society of Civil Engineers (ASCE) and the European Research Consortium confirmed that the PM formula can be used to obtain relatively accurate $ET_0$ calculations in both dry and humid climates [31]. Tan and Zheng used this model to simulate the $ET_c$ of seed corn and field corn crops in the Heihe River Basin, and the results were in line with the actual situation [17].

The PM formula is as follows:

$$ET_0 = \frac{0.408 \times \Delta \times (R_n - G) + \gamma \times 900/(T_a + 273) \times u_2 \times (e_s - e_a)}{\Delta + \gamma \times (1 + 0.34u_2)} \tag{2}$$

where $ET_0$ is the reference crop evapotranspiration (mm/day); $\Delta$ is the slope of the saturated vapor pressure vs. temperature curve (kPa/°C); $R_n$ is the net radiation of the reference crop canopy surface [MJ/(m²·day)]; $G$ is the soil heat flux [MJ/(m² day)]; $\gamma$ is the dry-wet table constant (kPa/°C); $T_a$ is the daily average temperature (°C) at a height of 2 m; $u_2$ is the wind speed (m/s) at a height of 2 m; $e_s$ is saturated water vapor, pressure difference (kPa); and $e_a$ is the actual vapor pressure (kPa). Among them, $\Delta$, $R_n$, $G$, $\gamma$, $e_s$, and $e_a$ can be calculated from the average temperature, average relative humidity, average wind speed, and sunshine hours.

### 3.2.2. Calculation of the $ET_c$ of the Crop

The CROPWAT model uses the single crop coefficient method to calculate $ET_c$, which is calculated as follows:

$$ET_c = K_c \times ET_0 \tag{3}$$

where $ET_0$ is the reference crop evapotranspiration (mm/day), $K_c$ is the crop coefficient at different growth stages, and $ET_c$ is the crop evapotranspiration (mm/day).

### 3.3. Improved Water Efficiency Model

On a regional scale, and in order to compare the differences in the WUE of different types of crops, Duan [21] proposed to convert the crop economic output into the market value according to its market value, so as to standardize the measurement of crop yield, and to characterize the WUE of crops with the economic benefit of unit water consumption. The calculation method is as follows:

$$WUE_1 = (Y \times P)/ET_c, \tag{4}$$

where $WUE_1$ is the water efficiency of the crop before the improvement, $Y$ (Yield) is the crop yield, $P$ (Price) is the crop market price, and $ET_c$ is the crop evapotranspiration.

Based on this method, this study improved the model and combined the actual input–output situation to remove the expenditures such as seeds, mulch, fertilizer, pesticide, machinery, and irrigation during each crop planting period, and characterize it with pure economic output value. The WUE of crops is calculated as follows:

$$WUE_2 = (Y \times P - C)/ET_c, \tag{5}$$

where $WUE_2$ is the improved crop water-use efficiency, $Y$ (Yield) is the crop yield, $P$ (Price) is the crop market price, $C$ (Cost) is the expenditure input, and $ET_c$ is the crop growth period evapotranspiration.

## 4. Results

### 4.1. Crop Input–Output Analysis

Using the input–output data of various crops in the questionnaire and following the value-based input–output model calculation, the main crop input–output in the study area was obtained (Table 3).

**Table 3.** Main crop input–output tables in the study area.

| Index / Crop | Input (Yuan/ha) | | | | | | | | Output | | | Income (Yuan/ha) |
| | Seed | Mulch Film | Pesticide | Fertilizer | Mechanical | Hire | Irrigation and Others | Input | Yield (kg/ha) | Unit Price (Yuan/kg) | Output (Yuan/ha) | |
|---|---|---|---|---|---|---|---|---|---|---|---|---|
| Wheat | 1890 | 0 | 375 | 2400 | 1560 | 0 | 1050 | 7275 | 6900 | 2.2 | 15,180 | 7905 |
| Barley | 1155 | 0 | 420 | 2850 | 1050 | 0 | 1050 | 6525 | 6940 | 2.0 | 13,880 | 7355 |
| Seed corn | 960 | 720 | 825 | 4425 | 1500 | 7800 | 1050 | 17,280 | 13,500 | 2.8 | 37,800 | 20,520 |
| Field corn | 990 | 630 | 555 | 5100 | 2025 | 6450 | 1050 | 16,800 | 14,250 | 2.0 | 28,500 | 11,700 |
| Rapeseed | 1200 | 0 | 825 | 2700 | 1050 | 0 | 1050 | 6825 | 3150 | 5.4 | 17,010 | 10,185 |
| Potato | 8250 | 900 | 1080 | 4620 | 2655 | 15,210 | 1050 | 33,765 | 37,500 | 1.2 | 45,000 | 11,235 |

From the input situation, the potato input per unit area is the highest, at 33,765 yuan/ha. The second is the production of seed corn and field corn, respectively, at 17,280 yuan/ha and 16,800 yuan/ha. Because wheat, rapeseed, and barley have no relatively low cost of mulch film and labor, they are 7275 yuan/ha, 6825 yuan/ha, and 6525 yuan/ha, respectively.

Due to the different yields and unit prices of different crops, the output is not the same. Among the six crops, potato has the highest unit output of 45,000 yuan/ha. The second is seed corn and field corn, the unit output being 37,800 yuan/ha and 28,500 yuan/ha, respectively. The yield of rapeseed among the six crops is the least, but because of its high unit price, the output is more than wheat and barley, at 17,010 yuan/ha. However, wheat and barley per unit area output are lower, less than 15,200 yuan/ha.

Finally, from the perspective of income, the yield per unit area of seed corn, which is 20,520 yuan/ha, is significantly higher than that of other crops, followed by field corn, which is 11,700 yuan/ha. Although potatoes have the most output, their corresponding investment is also the most, so the income after returning to two types of corn is 11,235 yuan/ha. The yield of rapeseed is not high, being 10,185 yuan/ha. The yield per unit area of wheat and barley is lower, amounting to less than 8000 yuan/ha.

Therefore, in general, when the planting conditions permit and only the net income is considered, the farmers should first choose to plant seed corn, followed by common corn and then by potato, rapeseed, wheat, and barley.

### 4.2. Characteristics of Crop Water Demand During the Growing Season

The $ET_c$ of the six major crops during the growing seasons in the study area was simulated using the CROPWAT model. Figure 3 shows the variation of $ET_c$ during the growth period of each crop. From Table 4 and Figure 3, we can see that the total $ET_c$ of field corn and seed corn, which is 597.2 mm and 577.3 mm, respectively, is higher than that of the other crops, and the $ET_c$ is the highest in July and August. The amount of rapeseed total $ET_c$ is 560.4 mm, and the maximum $ET_c$ is concentrated in June. The total $ET_c$ of wheat and barley is close, being 551.7 mm and 536.3 mm, respectively, and the $ET_c$ is mainly concentrated in May and June. The total $ET_c$ of potato is the lowest at 498.3 mm, and its peak $ET_c$ occurs in June.

**Table 4.** $ET_c$ of main crops in the study area (unit: mm).

| Crops / Month | March | April | May | June | July | August | September | October | Total Growth Period |
|---|---|---|---|---|---|---|---|---|---|
| Wheat | 13.1 | 49.9 | 145.7 | 229.7 | 113.3 | | | | 551.7 |
| Barley | 13.1 | 51.5 | 165.8 | 237.4 | 68.5 | | | | 536.3 |
| Seed corn | | 8.3 | 32.6 | 117.6 | 172.0 | 160.1 | 86.7 | | 577.3 |
| Field corn | | 6.1 | 25.4 | 103.8 | 169.5 | 162.8 | 116.4 | 13.2 | 597.2 |
| Rapeseed | | 36.0 | 92.4 | 194.0 | 180.4 | 57.6 | | | 560.4 |
| Potato | | 8.3 | 62.2 | 178.0 | 167.0 | 82.8 | | | 498.3 |

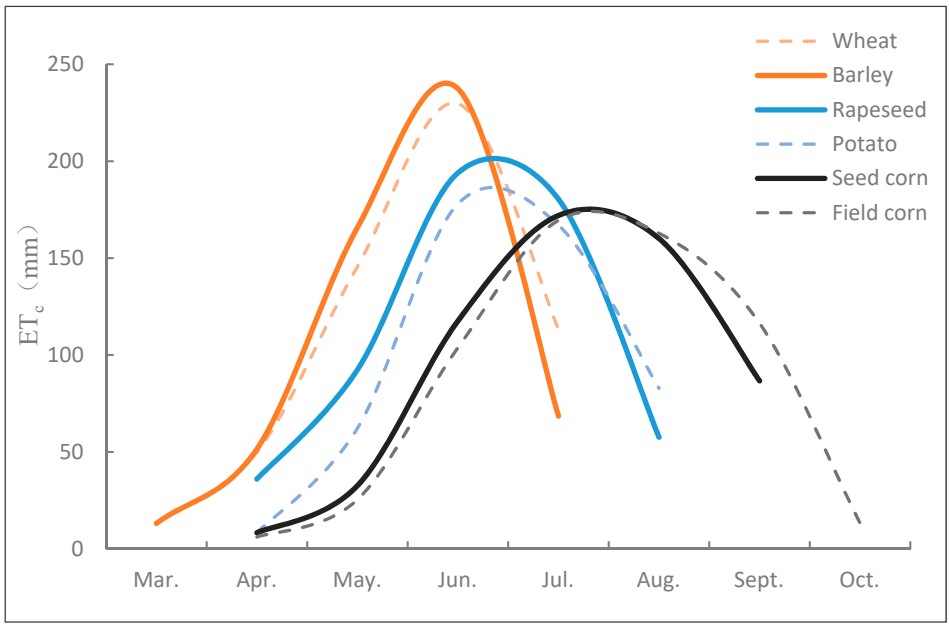

**Figure 3.** $ET_c$ of the crop growth period.

Therefore, in the case of planting conditions and only considering crop water saving, farmers are encouraged to first choose potato with the least water consumption, followed by barley, wheat, rapeseed, seed corn, and field corn.

### 4.3. Analysis of Crop WUE

Based on the per unit area yield, market price, production input, and growth evapotranspiration data for each crop, the WUE of the six crops in the study area was calculated based on Equations (4) and (5), respectively (Table 5, Figure 4).

**Table 5.** Water-use efficiency (WUE) of the main crops in the study area.

| Crops | Index | | | | | |
|---|---|---|---|---|---|---|
| | Yield (Y) (kg/ha) | Price (P) (Yuan/kg) | Input (C) (Yuan/ha) | $ET_c$ (mm) | WUE$_1$ (Yuan/m$^3$) | WUE$_2$ (Yuan/m$^3$) |
| Wheat | 6900 | 2.2 | 7275 | 551.7 | 2.75 | 1.43 |
| Barley | 6940 | 2.0 | 6525 | 536.3 | 2.59 | 1.37 |
| Seed corn | 14,250 | 2.0 | 16,800 | 577.3 | 4.94 | 2.03 |
| Field corn | 13,500 | 2.8 | 17,280 | 597.2 | 6.33 | 3.44 |
| Rapeseed | 3150 | 5.4 | 6825 | 560.4 | 3.04 | 1.82 |
| Potato | 37,500 | 1.2 | 33,765 | 498.3 | 9.03 | 2.25 |

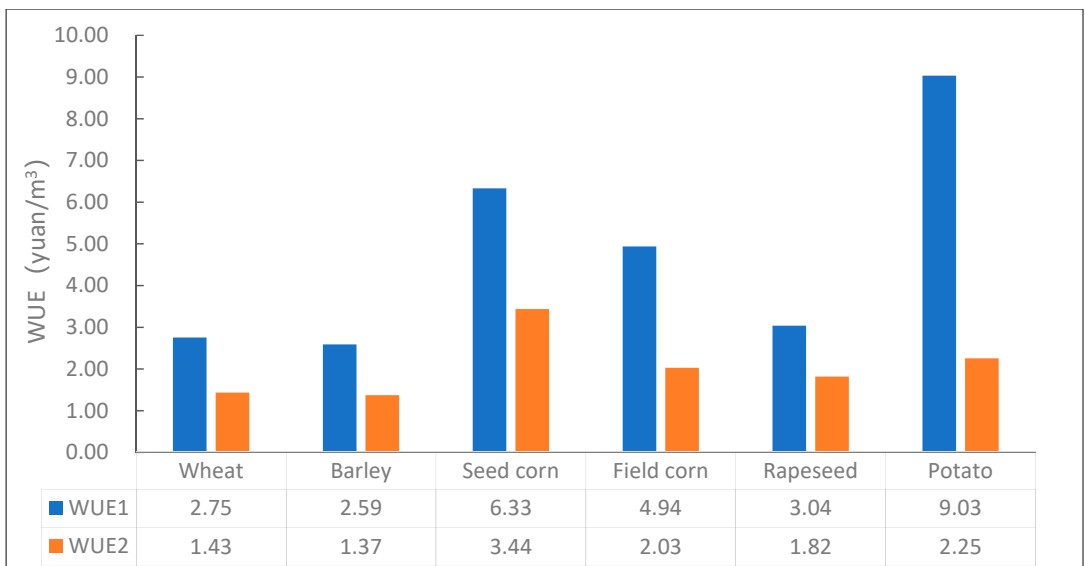

**Figure 4.** WUE of the crops in the study area.

Regardless of the cost input, and using the model (4) calculation, potato has the highest water efficiency, up to 9.03 yuan/m$^3$, followed by seed corn at 6.33 yuan/m$^3$. Then follow field corn, rapeseed, and wheat. Barley has the lowest WUE of only 2.59 yuan/m$^3$.

Considering cost input, the WUE rank of the six crops calculated using model (5) is as follows: seed corn, potato, field corn, rapeseed, wheat, and barley.

Through comparative analysis, we can see that the WUE of the crops changed when considering cost. Among them, the WUE of seed corn became the highest, at 3.44 yuan/m$^3$, whereas the WUE of potato was reduced to 2.25 yuan/m$^3$. The WUE rank of the other crops was as follows: field corn, rapeseed, wheat, and barley. Therefore, when considering the economic benefits of crops and saving water, farmers are encouraged to give priority to planting seed corn, followed by potato, then field corn, rapeseed, wheat, and barley.

## 5. Conclusions and Discussion

### 5.1. Conclusions

In this paper, the input–output model was used to calculate the input and output of crops. The CROPWAT model was used to estimate the main crop evapotranspiration in the study area, and the water demand characteristics of the main crops in the Zhangye area were analyzed. Then, the crop water efficiency model was improved and the differences in WUE of the main crops before and after improvement were compared and analyzed. The following conclusions can be drawn:

(1) With regard to the input–output table of crops, from the output perspective, potato has the highest unit output of 45,000 yuan/ha among the six crops, followed by seed corn, which is 37,800 yuan/ha, and then by field corn, rapeseed, wheat, and barley. In terms of income, the yield per unit area of seed corn is 20,520 yuan/ha, followed by field corn, which is 11,700 yuan/ha. Potato's yield per unit area is 11,235 yuan/ha. In the case of pure income only, farmers should first choose to plant seed corn, followed by field corn, followed by potato, rapeseed, wheat, and barley.

(2) As for the $ET_c$ of crops, the $ET_c$ of field corn and seed corn, which is 597.2 mm and 577.3 mm, respectively, is higher than the other crops. The $ET_c$ of rapeseed is 560.4 mm. The $ET_c$ of wheat and barley is relatively close, being 551.7 mm and 536.3 mm, respectively. Finally, the $ET_c$ of potato is the lowest, which is 498.3 mm. Therefore, in the case of only considering water conservation, farmers are encouraged to first choose to plant potato with the least amount of water, followed by barley, wheat, rapeseed, seed corn, and field corn.

(3) Regarding WUE, it was calculated using the improved model (5). Compared with the results calculated using model (4), the WUE of the crop changes. Among them, the WUE of seed corn becomes the highest reaching 3.44 yuan/m$^3$, whereas the WUE of potato drops to 2.25 yuan/m$^3$. Then, the WUE rank of the crops is field corn, rapeseed, wheat, and barley. When considering the economic benefits of crops and saving water, farmers are encouraged to give priority to planting seed corn, followed by potato, field corn, rapeseed, wheat, and barley.

*5.2. Discussion*

In this study, the WUE of the six main crops in Zhangye City was calculated by improving the water efficiency model. Considering both income and water conservation, this paper puts forward the following suggestions for crop structure adjustment in arid regions. Since the pure economic benefit generated by the unit water consumption of corn is the highest, it can be vigorously promoted in arid areas where water is scarce. This can not only make full use of irrigation water, but also protect the economic income of farmers. Potato has a high input cost, but because of its own low water consumption and high yield, its WUE is relatively high, and it can be planted in arid areas. Field corn has a high input cost and its own water consumption is high, so large-scale planting in arid areas is not profitable. Rapeseed has a low unit price, and it can be promoted when conditions permit and developed into local tourism agriculture.

Of course, in arid regions, farmers are encouraged to plant crops, in addition to considering crop economic benefits and water consumption, but also to consider other influencing factors, such as natural conditions (precipitation, temperature, accumulated temperature, etc.), land quality, population factors (population structure, population education level, etc.), policy factors, among others. Therefore, further research and discussion is required to discover how to adjust the agricultural structure to its optimal level in order to use the agricultural resources in an arid area efficiently and continuously.

**Author Contributions:** X.X. came up with the idea, designed the research, improved the model of WUE, wrote the manuscript and reviewed it. L.F. collected and analyzed the data and wrote the manuscript. X.L. and T.J. supervised the research and gave scientific advice. M.T. provided a part of data and reviewed the paper. L.Z. ran the CROPWAT model. F.J. collected and analyzed the data.

**Funding:** This work was supported by the National Natural Science Foundation of China (Grant No: 91325302). This paper funded by the Domestic Visiting Scholar Program for Young Backbone Teachers of Colleges and Universities in Shandong Province.

**Conflicts of Interest:** The authors declare no conflict of interest.

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
