# Peer review of "Water-Use Efficiency of Crops in the Arid Area of the Middle Reaches of the Heihe River: Taking Zhangye City as an Example"

_water, doi:10.3390/w11081541_

Round 1
Reviewer 1 Report
The English language needs a careful edit I have made a lot of changes but probably missed some and left some that were acceptable but could be improved. My main concern is that there is no sensitivity analysis for the outcomes and recommendations. To say one crop is better than another based on a single point value is a bit of a stretch. There should be data on mean input price and output price variability over the last few years. Similarly on yields and on season length and conditions. thus ranges can be generated for the profit and it would be very interesting to see where these ranges lie. I believe the authors would have a tremendously improved paper with added sensitivity analysis.
Author Response
Dear Editors and Reviewers:
Please see the attachment.

Reviewer 2 Report
Dear Authors, you should address all my recommendations reported across the text, tables and figures.

Author Response

(The authors gave the same response as above.)

Round 2
Reviewer 2 Report
Dear Authors, my recommendations have been addressed and therefore the manuscript can be accepted for publication in this Journal in my opinion.
This manuscript is a resubmission of an earlier submission. The following is a list of the peer review reports and author responses from that submission.